# From CENTRAL to SENTRAL (SErum aNgiogenesis cenTRAL): Circulating Predictive Biomarkers to Anti-VEGFR Therapy

**DOI:** 10.3390/cancers12051330

**Published:** 2020-05-22

**Authors:** Riccardo Giampieri, Pina Ziranu, Bruno Daniele, Antonio Zizzi, Daris Ferrari, Sara Lonardi, Alberto Zaniboni, Luigi Cavanna, Gerardo Rosati, Mariaelena Casagrande, Nicoletta Pella, Laura Demurtas, Maria Giulia Zampino, Pietro Sozzi, Valeria Pusceddu, Domenico Germano, Eleonora Lai, Vittorina Zagonel, Carla Codecà, Michela Libertini, Marco Puzzoni, Roberto Labianca, Stefano Cascinu, Mario Scartozzi

**Affiliations:** 1Medical Oncology Unit, University Hospital and Università Politecnica delle Marche, 60126 Ancona, Italy; riccardo.giampieri81@gmail.com (R.G.); antonio.zizzi@ospedaliriuniti.marche.it (A.Z.); 2Medical Oncology Unit, University Hospital and University of Cagliari, 09042 Monserrato, Italy; pi.ziranu@gmail.com (P.Z.); lau.demi@tiscali.it (L.D.); valeria.pusce@gmail.com (V.P.); ele.lai87@gmail.com (E.L.); marcopuzzoni@gmail.com (M.P.); 3Medical Oncology Unit, Ospedale G. Rummo, 82100 Benevento, Italy; b.daniele@libero.it (B.D.); domgerm@libero.it (D.G.); 4Medical Oncology Unit, Azienda Ospedaliera San Paolo, 20142 Milano, Italy; daris.ferrari@asst-santipaolocarlo.it (D.F.); carla.codeca@ao-sanpaolo.it (C.C.); 5Medical Oncology Unit 1, Istituto Oncologico Veneto-IRCCS, 35128 Padova, Italy; sara.lonardi@iov.veneto.it (S.L.); vittorina.zagonel@iov.veneto.it (V.Z.); 6Medical Oncology Unit, Fondazione Poliambulanza, 25124 Brescia, Italy; alberto.zaniboni@poliambulanza.it (A.Z.); michela.libertini@poliambulanza.it (M.L.); 7Medical Oncology Unit, Ospedale di Piacenza, 29121 Piacenza Italy; l.cavanna@ausl.pc.it; 8Medical Oncology Unit, Ospedale San Carlo, 85100 Potenza, Italy; oncogerry@yahoo.it; 9Medical Oncology Unit, Azienda Ospedaliero Universitaria Santa Maria della Misericordia, 33100 Udine, Italy; mariaelena.casagrande@asuiud.sanita.fvg.it (M.C.); nicoletta.pella@aoud.sanita.fvg.it (N.P.); 10Gastrointestinal Medical Oncology Unit and Neuroendocrine Tumors, Istituto Europeo di Oncologia-IRCCS, 20141 Milano, Italy; maria.zampino@ieo.it; 11Medical Oncology Unit, Nuovo Ospedale degli Infermi, 13900 Biella, Italy; pietrosozzi@libero.it; 12Medical Oncology Unit, ASST Papa Giovanni XXIII, 24127 Bergamo, Italy; rlabian@tin.it; 13IRCCS San Raffaele Scientific Institute Hospital, 20132 Milan, Italy; cascinu.stefano@hsr.it

**Keywords:** colon cancer, bevacizumab, FGF2, PlGF, VEGF, angiogenesis, circulating biomarkers

## Abstract

Background: In the last decade, a series of analyses failed to identify predictive biomarkers of resistance/susceptibility for anti-angiogenic drugs in metastatic colorectal cancer (mCRC). We conducted an exploratory preplanned analysis of serum pro-angiogenic factors (SErum aNgiogenesis-cenTRAL) in 72 mCRC patients enrolled in the phase II CENTRAL (ColorEctalavastiNTRiAlLdh) trial, with the aim to identify potential predictive factors for sensitivity/resistance to first line folinic acid-fluorouracil-irinotecan regimen (FOLFIRI) plus bevacizumab. Methods: First-line FOLFIRI/bevacizumab patients were prospectively assessed for the following circulating pro-angiogenic factors, evaluated with ELISA (enzyme-linked immunosorbent assay)-based technique at baseline and at every cycle: Vascular endothelial growth factor A (VEGF-A), hepatocyte growth factor (HGF), stromal derived factor-1 (SDF-1), placental derived growth factor (PlGF), fibroblast growth factor-2 (FGF-2), monocyte chemotactic protein-3 (MCP-3), interleukin-8 (IL-8). Results: Changes in circulating FGF-2 levels among different blood samples seemed to correlate with clinical outcome. Patients who experienced an increase in FGF-2 levels at the second cycle of chemotherapy compared to baseline, had a median Progression Free Survival (mPFS) of 12.85 vs. 7.57 months (Hazard Ratio—HR: 0.73, 95% Confidence Interval—CI: 0.43-1.27, *p* = 0.23). Similar results were seen when comparing FGF-2 concentrations between baseline and eight-week time point (mPFS 12.98 vs. 8.00 months, HR: 0.78, 95% CI: 0.46–1.33, *p* = 0.35). Conclusions: Our pre-planned, prospective analysis suggests that circulating FGF-2 levels’ early increase could be used as a marker to identify patients who are more likely to gain benefit from FOLFIRI/bevacizumab first-line therapy.

## 1. Introduction

The chimeric monoclonal immunoglobulin G3 (IgG3) antibody directed against soluble circulating vascular endothelial growth factor-A (VEGF-A), bevacizumab, gained a prominent role in different solid tumor types, such as in breast, ovarian, kidney, and lung cancers [1,2,3,4] since its introduction as a first line therapeutic option for metastatic colorectal cancer (mCRC) patients [5]. On the other hand, adjuvant trials in resected colorectal cancer (CRC) patients unexpectedly failed their primary endpoint [6,7,8]. The lack of information regarding the in vivo activity of bevacizumab and tumor microenvironment changes during treatment represents a relevant issue in this setting.

Despite the highly selective mechanism of action, no reliable predictive markers have been identified for bevacizumab activity. In particular, no validated biomarkers (baseline VEGF-A circulating serum levels, their changes during treatment, VEGF-A activating mutations, and VEGF-A polymorphisms or its principal receptors, VEGFR-1 and VEGFR-2), seem to be able to predict the treatment response probability.

Although VEGF-A represents one of the most active factors responsible for tumor-driven angiogenic switch, other pro-angiogenic factors may have a crucial role in the maintenance of preexisting tumor blood vessels and in the neo-angiogenesis process induction. As a consequence, resistance to the VEGF-A inhibition may be extremely challenging to prove. According to preclinical data, clinical progression during bevacizumab treatment is likely to be induced by the secondary neo-angiogenesis processes activation rather than by a single molecular change. This biological process would maintain blood supply to the tumor, allowing escape from VEGF-A blockade [9].

In this scenario, different pro-angiogenic factors could be responsible for a quick escape from VEGF inhibition, more specifically, stromal derived factors (such as the stromal derived factor-1, SDF-1), chronic inflammatory factors (such as interleukin-8, the monocyte chemotactic protein-3 (MCP-3), cells or tissue growth factors implied in the neo-angiogenesis processes, such as the hepatocyte growth factor (HGF), placental derived growth factor (PlGF), fibroblast growth factor-2 (FGF-2).

In particular, several preclinical studies demonstrated that the angiogenesis blockade in tumor-bearing mice upregulates expression of PlGF, VEGF, FGF-2, and other angiogenic factors [10,11]. Furthermore, in glioblastoma patients, plasma levels of FGF-2 and SDF-1 care increased upon disease progression under VEGF-targeted therapy [12]. Even in glioblastoma patients, HGF/tyrosine protein kinase Met (c-MET) pathway was highly upregulated during the recurrence after bevacizumab treatment. This phenomenon is hypoxia-dependent and it has not been recorded in patients without anti-VEGF treatment [13]. MCP-3 is often produced by tumor cell lines. Its production might contribute to cancer cells’ invasion and metastasis, regulating protease secretion through macrophages [14]. Finally, some inflammatory cytokines might have potent pro-angiogenic effects. In particular, interleukin-8 (IL-8) is a pro-angiogenic factor produced by tumor-infiltrating macrophages that has been revealed to facilitate the development of angiogenesis in several cancers. In xenograft cancer models, anti-angiogenic resistance coincided with increased secretion of IL-8 from tumor cells to the plasma [15].

The release of these angiogenic factors (VEGF-A, HGF, SDF1, PlGF, FGF-2, MCP-3, and IL-8), induced by hypoxia in the tumor tissue may be associated with anti-VEGF resistance. The choice to monitor their serum levels can be a surrogate predictive biomarker of anti-angiogenic therapies.

Other pro-angiogenic factors work as alternative ligands for VEGFR-2 and, if upregulated, they could represent mechanisms of resistance to anti-angiogenics. In particular, vascular endothelial growth factor-C (VEGF-C) and vascular endothelial growth factor-D (VEGF-D) are involved in lymphangiogenesis, due to the high affinity for vascular endothelial growth factor receptor-3 (VEGFR-3) expressed on lymphatic endothelial cells [16]. Their poor affinity for VEGFR-2 activates the angiogenesis occasionally. To date, their role in angiogenesis remains controversial. The literature data support that VEGF-C has been associated with angiogenesis in breast cancer showing its cooperation with FGF-2 and VEGF-A in order to induct angiogenesis. Another study suggested that VEGF-C induces blood vessel changes without evidence of new angiogenesis [17,18,19]. Few data reports comment on the role of VEGF-D and angiogenesis, but a study over CRC patients found that lower expression of VEGF-D was associated with an outcome benefit from bevacizumab treatment [20,21].

In our study, we selected pro-angiogenic biomarkers with particular interest on those that showed more correlations with anti-angiogenic drugs’ resistance.

We conducted an exploratory pre-planned analysis (SErum aNgiogenesis-cenTRAL) in mCRC patients enrolled in the phase II CENTRAL (ColorEctalavastiNTRiAlLdh) trial of first-line folinic acid-fluorouracil-irinotecan regimen (FOLFIRI) and bevacizumab, who had been prospectively stratified according to serum lactate dehydrogenase (LDH) value [22]. Our aim was to identify concentration changes of circulating pro-angiogenic factors during treatment as a potential predictive factor for efficacy/resistance to FOLFIRI/bevacizumab treatment.

## 2. Results

The survival analysis refers to data collected up to April 2016, at approximately three years since the study start and one year after the CENTRAL study conclusion [22]. Median progression-free survival (mPFS) was 12.98 months and median overall survival (mOS) was 24.52 months.

Blood sample A was available in 72 patients, whereas blood sample B and C were obtained in 71 and 66 patients, respectively. Failure to provide a blood sample was mainly due to early treatment discontinuation: Two patients received heart failure diagnoses before radiological evaluation, one patient died during treatment for causes other than cancer, three patients developed severe toxicities (≥ grade 3 according to National Cancer Institute-Common Terminology Criteria for Adverse; NCI-CTCAE G3), four patients stopped treatment after radiological response at their will, two patients stopped treatment after radiological response to undergo metastases surgery, and, finally, two patients continued treatment in different Oncology Departments, exiting the study.

### 2.1. Serum Biomarkers’ Analysis

No significant correlation was found between plasma levels and clinical outcome for all biomarkers analyzed, except for FGF-2.

We found a high inter-patient variability for baseline FGF-2 plasma levels. Blood sample A median FGF-2 serum level was 44.963 picograms per millilitre (pg/mL) (range: 13.203–73.618) and mean FGF-2 serum level was 46.355 pg/mL (standard deviation, SD: 11.060). Inter-patient variability was greater for blood sample B, too, with a median FGF-2 serum level of 44.919 (range 12.537–72) and mean FGF-2 serum level of 47.103 (SD: 11.763). Similarly, in blood sample C, FGF2 levels showed high inter-patient variability, median FGF-2 serum level was 43.509 (range 0.544–78.735), and mean FGF2 serum level was 44.582 (SD: 14.141) (Figure 1).

#### 2.1.1. Analysis for Sample A FGF-2 Levels

No outcome differences were found considering different baseline FGF-2 concentrations. In particular, in patients with higher vs. patient with lower median FGF-2 level the mPFS was, respectively, 8.52 vs. 8.60 months (HR: 1.16, 95% CI: 0.70-1.92, *p* = 0.53) (Figure 2a) and mOS was, respectively, 24.52 vs. 25.47 months (HR: 1.70, 95% CI: 0.68-4.30, *p* = 0.26). Furthermore, not-significant difference was observed in term of response rate (RR) (40% vs. 34%, *p* = 0.63) or progression disease (PD) (16% vs.11%, *p* = 0.73). Stratifying patients in quartiles (first quartile: <40.945 pg/mL, second quartile: 40.945–44.964 pg/mL, third quartile: 44.945–50.819 pg/mL, fourth quartile >50.819 pg/mL), no statistically significant differences were seen among the four groups in term of PFS (Figure 2b) and OS. Patients included in the fourth quartile had a trend towards worse OS (mOS, respectively, not yet reached (NR) vs. 24.85 vs.NR vs. 20.75 months, for first vs. second vs. third vs. fourth quartile, *p* = 0.47).

#### 2.1.2. Analysis for Differences in FGF-2 Concentrations between Samples A and B

In 71 patients, comparison of FGF-2 concentrations between sample A/B was available. We observed a median ratio of 105%, with a range of 34.85%–154.24% (SD: 21.31, with a normal distribution as D’agostino-Pearson test for normal distribution with *p* = 0.09). A trend towards an increase of the concentration of FGF-2 levels from sample A to sample B was seen.

We compared survival outcomes and Response Rate-Disease Control Rate (RR-DCR), stratifying patients on the basis of the reduction (<100% from sample A–B) or the increase (>100% from sample A–B) of FGF-2 levels. An increase was observed in 44 patients (62%) and a reduction in 37 (58%) patients.

In patients with FGF-2 levels increased between sample A/B, there was a trend towards better outcomes. Indeed, in patients with FGF-2 level increased vs. decreased, the mPFS was, respectively, 12.85 vs. 7.57 months (HR: 0.73, 95% CI: 0.43–1.27, *p* = 0.23) (Figure 3a) and mOS was, respectively, 25.47 vs. NR, (HR: 0.59, 95% CI: 0.22–1.59, *p* = 0.22). No significant difference was observed in terms of RR (34% vs. 44%, *p* = 0.45) and PD (13% vs. 15%, *p* = 1).

Stratifying patients by different percentile change of FGF-2 concentrations, the 10th percentile was 80%, the 25th percentile was 90%, the 75th percentile was 114%, and the 90th percentile was 123%. We identified three groups (<90%, 90%–114%, >114% ratio between A/B) using as cutoff the 25th and 75th percentile reduction/increase values in order to have an adequate patient number to analyze. Among these three different groups, a better mPFS (respectively, 6.95 vs. 8.49 vs. 14.66 months, *p* = 0.32) (Figure 3b) and a better mOS (20.75 vs. 25.47 months vs. NR, *p* = 0.35) were observed in patients with increased concentrations, although not statistically significant. Furthermore, RR was not significantly different among these three groups (46% vs. 37% vs. 33%, *p* = 0.71) as well as PD (20% vs. 13% vs. 11%, *p* = 0.74).

#### 2.1.3. Analysis for Differences in Concentrations of FGF-2 between Samples A and C

In 66 patients, comparison of FGF-2 concentrations between sample A/C was available. We observed a median ratio of 97.03%, with a range 3.36%–190.93% (SD: 31.98, with a normal distribution as D’agostino-Pearson test for normal distribution with *p* = 0.18).

We compared survival outcomes and RR-DCR, stratifying patients on the reduction (<100% from sample A–C) or the increase (>100% from sample A–C) of FGF-2 levels. An increase was observed in 33 patients (50%) and a reduction in 33 (50%) patients.

There was a better mPFS trend in patients with FGF-2 levels increased compared with decreased between sample A/C (respectively, 12.98 vs. 8.00, HR: 0.78, 95% CI: 0.46–1.33, *p* = 0.35) (Figure 4a). Instead, mOS was not significantly different in patients experiencing FGF2 concentrations increase/decrease (respectively, 25.47 vs. 24.52, HR: 0.58, 95% CI: 0.22-1.35, *p* = 0.27). Similarly, RR (42% vs. 33%, *p* = 0.61) and PD (6% vs. 15%, *p* = 0.42) were not significantly different.

Stratifying patients by different percentile change of FGF-2 concentrations, the 10th percentile was 59%, the 25^th^ percentile was 79%, the 75th percentile was 118%, and the 90th percentile was 137%. We identified three groups (<79%, 79%–118%, >118% ratio between A/C) using as cutoff the 25th and 75th percentiles’ reduction/increase values in order to have an adequate patient number to analyze. Among these three different groups, a better mPFS (respectively, 8.00 vs. 12.85 vs. 18.91 months, *p* = 0.35) (Figure 4b) and better mOS (respectively, 20.75 vs. NR vs. 25.47, *p* = 0.25) were observed in patients with increased concentrations, although not statistically significant. Furthermore, RR was not significantly different among the three groups (43% vs. 33% vs. 41%, *p* = 0.73), as well as PD (18% vs. 10% vs. 6%, *p* = 0.44).

Finally, considering FGF-2 levels’ changes through the various time points, an overall FGF-2 circulating blood levels’ increasing trend after eight weeks of treatment was observed, both as terms of FGF-2 concentration and percentage change from baseline levels (Figure 5 and Figure 6, respectively).

#### 2.1.4. Other Analyses Concerning Circulating FGF-2 among Different Time Points and Clinical Factors

The clinical characteristics of the population included in this study can be found in Table 1.

Baseline Performance Status 1, according to Eastern Cooperative Oncology Group scale, (ECOG PS-1) patients had a worse mPFS (6.13 vs. 10.56 months, HR: 2.23, 95% CI: 0.96–5.17, *p* = 0.0091) compared to ECOG-PS0 patients. Furthermore, patients with symptomatic disease (ECOG-PS1) with FGF-2 levels increased between baseline and eight weeks of treatment had a statistically significantly improved mPFS (6.13 vs. 2.52 months, HR: 3.67, *p* = 0.0294) (Figure 7). Interestingly, mPFS was not statistically different according to tumor sidedness (respectively, 10.57 vs. 8.53 months left vs. right, HR: 0.95, 95% CI: 0.57–1.59, *p* = 0.85). V-Raf murine sarcoma viral oncogene homolog B *(B-RAF)* and Neuroblastoma RAS viral oncogene homolog *(N-RAS)* mutant patients had shorter PFS than Kirsten rat sarcoma 2 viral oncogene homolog *(K-RAS)* mutant and wild-type patients (6.62 vs. 8.62, HR: 2.38, 95% CI: 0.41–13.57, *p* = 0.12); however, the small number of patients (four *B-RAF* mutated and one *N-RAS* mutated) reduced the statistical significance.

VEGF-A serum levels reduced in 58% of the patients, without any significant correlation with clinical outcome. We did not observe any difference in terms of PFS between a decrease vs. an increase in VEGF-A levels (mPFS 14.16 vs. 12.85 months, HR: 0.95, 95% CI: 0.51–1.76, *p* = 0.87), either, in OS (mOS NR vs. 24.85 months, HR: 0.87, 95% CI: 0.34–2.18, *p* = 0.76). Changes in VEGF-A values were not associated with levels of circulating FGF2 during treatment (*p* = 0.44).

## 3. Discussion

Bevacizumab-based chemotherapy regimens rapidly developed as a mainstay of mCRC treatment worldwide [23,24,25,26]. Indeed, the molecular mechanisms underlying tumor-induced neo-angiogenesis [27,28,29,30] represent a crucial therapeutic target for these patients.

The biological complexity of these pathways translates into the challenging reliable criteria selection for patients who are more likely to benefit from this therapeutic approach. A series of analyses tried to identify resistance/susceptibility predictive factors for anti-angiogenic drugs, failing to confirm findings with immediate implications for the daily clinical practice. Among them, VEGF polymorphisms and related pathways have been assessed as resistance factor in a series of papers. Loupakis et al. [29] retrospectively investigated the VEGF polymorphisms’ impact during first-line FOLFIRI/bevacizumab treatment in mCRC patients. The VEGF rs833061 polymorphism seemed associated to worse prognosis for patients with T/T genotype (HR for PFS: 2.13, 95% CI: 1.41–5.10, *p* = 0.0027). Similarly, VEGF-A rs2010963 polymorphism correlated with better outcome for heavily pretreated mCRC patients receiving regorafenib [23]. Despite these promising results, the prospective validations of these results are lacking or failed [30]. Probably, polymorphisms are unlikely to be able to define the ever-changing neo-angiogenesis process. Otherwise, the possibility to assess tumor angiogenesis dynamics recently emerged as a potentially more effective way to investigate this particular aspect of tumor biology.

In our study, the circulating FGF-2 serum levels may correlate with clinical outcome for patients treated with FOLFIRI/bevacizumab. In particular, there were no differences in terms of mPFS, mOS, and RR in patients with higher vs. lower median FGF-2 level. However, after patients’ stratification into quartiles, although not statistically significant, those included in the fourth quartile showed a worse trend in OS. Therefore, while low baseline circulating FGF-2 levels seem to suggest an improved prognosis, the increase of FGF-2 levels along the treatment might be associated with better survival outcomes. Furthermore, FGF-2 levels continued to increase during the course of treatment, reaching their highest value shortly before progression.

Probably, it could be explained by the FGF-2 pathway activation as a mechanism of resistance to anti-angiogenic therapy. Therefore, a blockade of VEGF-A mediated by bevacizumab may induce an immediate activation of FGF-2, avoiding anti-angiogenic treatment.

Several studies showed the prognostic role of circulating pro-angiogenic factor in CRC. Kwon et al. analyzed serum samples from 132 CRC patients undergoing curative resection. They showed that high preoperative VEGF levels were associated with increased tumour size and higher Carcinoembryonic antigen (CEA) levels. In a multivariate analysis, higher VEGF-A was an independent prognostic factor for shorter OS (HR: 4.779, 95% CI: 1.15–19.94, *p* = 0.032) [31]. Afterwards, in a retrospective subset analysis over more than 2000 mCRC patients, higher baseline levels of serum/plasma VEGF-A and CEA were confirmed as prognostic biomarkers for poorer PFS and OS in patients with mCRC, with no treatment correlation [32].

Similar results analyzing different circulating pro-angiogenic factors (PlGF and VEGF-D) were demonstrated by Lieu et al. [20]. The authors considered two different retrospectively assessed cohorts of mCRC patients, one treated with bevacizumab-based regimens and the other without anti-angiogenic drugs. During treatment, PlGF and VEGF-D plasma concentrations reached their highest value (PlGF increase of 43% and VEGF-D increase of 6% from baseline) just before PD. This increase was transient in time, reverting back to lower levels after six weeks since treatment withdrawal.

Likewise in our analysis, a circulating increase of PlGF levels during treatment was demonstrated (data not shown), with a median increase of 40% (baseline median PlGF: 21.88 pg/mL, before PD: 30.653 pg/mL) in PlGF levels from baseline to progression. However, we could not find any significant correlation between PlGF levels’ increase and clinical outcome.

These analyses suggest that patients receiving bevacizumab treatment may show pro-angiogenic factors’ increase that potentially explains the secondary resistance. Among these factors, FGF2 represents one of the most interesting in terms of reproducibility. Dosing FGF2 serum levels may provide an indirect, non-invasive way to monitor cancer progression.

The role of FGF-2 in acquired resistance to anti-VEGF-based therapy was already discussed in some preclinical studies. Gyanchadani et al. demonstrated that, in head and neck cancer (HNC) cell lines, acquired resistance to Bevacizumab treatment is associated with upregulation of different pro-angiogenic factors. Among them, FGF-2 plays a crucial role and has been found to be associated with restored sensitivity to bevacizumab when inhibition of FGF-2 was performed in their model [33]. Ichikawa et al. reported that in cell lines and in an animal model that acquired resistance to anti-VEGF-based therapy is associated with a similar over-expression of both FGF-2 and vascular endothelial growth factor receptor-2 (FGFR-2). Tumor over-expressing the mouse VEGFR2 extracellular domain fused with the human IgG4 fragment crystallizable (Fc) region (VEGFR-2-Fc) showed highest levels of mRNA expression from FGF2 and other factors of FGF-2/FGFR-2 pathway [34]. Finally, in ovarian cancer resistant to bevacizumab, Guerrouehan et al. demonstrated that Protein Kinase B (PKB also known as AKT)-mediated endothelial factors secretion (where FGF-2 plays a major role) creates an autocrine loop that avoids bevacizumab inhibition. Cell lines subject to this study were obtained through a continuous exposure to bevacizumab, in order to facilitate the resistance model acquisition [35].

Conversely, the prognostic impact of circulating FGF-2 levels is still controversial [36,37,38,39,40]. Madsen et al. [36] reported the outcome of ovarian cancer patients treated with bevacizumab, stratified according to different pro-angiogenic factors’ levels, including FGF-2. The authors suggested that FGF-2 baseline levels might have a negative prognostic role. Patients having concentrations higher than 75th percentile of the distribution showed a shorter PFS compared to other subgroups. Furthermore, in patients who had PD along the treatment, a FGF-2 levels’ decrease was observed. These results seem consistent with our data, showing that patients with FGF-2 levels’ increase during treatment had a longer PFS and OS, despite the negative prognostic value of higher baseline FGF-2 levels.

A clear explanation to this biological phenomenon is difficult to find. A FGF-2 levels’ decrease might be related to an ineffective inhibition of VEGF-A. Consequently, tumor cells would not induce the other pro-angiogenic factors’ production because VEGF-A is already sufficient to sustain the neo-angiogenetic process. The only caveat against this explanation is that our analysis failed to demonstrate a correlation between VEGF-A and clinical outcome. These results were in concordance with those presented by Madsen et al. who showed a VEGF-A circulating levels’ reduction independently by the response profile experienced during treatment [36]. Our analysis seems to contradict the hypothesis of a correlation between VEGF levels and FGF-2.

Notably, most of these pro-angiogenic factors (VEGF-A and PlGF) are stored inside platelets. Platelet-derived angiogenic cytokines are released from activated platelets, from ethylenediamine tetraacetic acid (EDTA)-activated platelets, in particular. There is a large variation susceptibility to platelet EDTA activation among individuals. Thus, their release by platelets could reduce the reliability of their own circulating levels’ assessment as markers of different anti-angiogenic activity [41,42]. Conversely, FGF2 is released by endothelial cells, reducing the confounding factor of platelet-poor plasma dropout during blood samples’ processing.

In addition, increased FGF-2 levels have been found in several human cancers, but FGF-2 levels do not always correlate with microvessel density [43,44]. In fact, FGF-2 contributes to tumor progression not only by inducing neovascularization but also by modulating growth, differentiation, migration, and survival in a variety of cancer cell types [9,45]. Several studies have shown that FGF-2 is a key cancer-promoting factor in the tumor microenvironment, regulating cross-talk between epithelial and stromal tumor compartments. In particular, FGF-2 expression is high in the tumor stroma, including inflammatory cells, myofibroblasts, and endothelial cell, suggesting that FGF-2 can modulate tumor progression [46]. This may partially explain the difficulty in demonstrating the prognostic role of FGF-2 in preclinical studies. Cancer cell lines in vitro and in vivo trial do not totally represent microenvironment heterogeneity and complexity [47,48].

Moreover, the worse prognosis of the patients with FGF2 levels’ decrease might be related to angiogenesis inhibition that is not “per se” sufficient to halt tumor progression.

Some preclinical data suggest a potential synergistic effect between bevacizumab and chemotherapy agents [49]. Other studies support the imperfect drug penetration hypothesis, caused by angiogenesis inhibition. This may lead to single drug exposure of the affected tumor tissue (effective spatial monotherapy) [50]. In particular, this factor probably determines a tumor cell exposure reduction versus chemotherapy in those patients who have larger tumor burden, speeding up chemotherapy resistance development during treatment [51].

## 4. Patients and Methods

The University Hospital “Ospedali Riuniti” of Ancona Ethical Committee approved the analysis. This study was performed in accordance with the study protocol and the ethical principles stated in the Declaration of Helsinki as well as those indicated in the International Conference on Harmonization (ICH) Note for Guidance on Good Clinical Practice (GCP, ICH E6, 1995) and all applicable regulatory requirements. All patients had to sign a written informed consent before study entry. Adequate information was given to eligible patients by the principal investigator or co-investigators at each participating Center and in accordance with local regulations. Written informed consent to participate in the clinical study had to be given before any study-related activities were carried out. The declaration of informed consent was personally signed and dated by the subject, and by the investigator/person designated by the investigator to conduct the informed consent discussion.

### 4.1. Patients

All patients receiving FOLFIRI/bevacizumab within the CENTRAL trial were eligible for this analysis (Table 1) [22].

### 4.2. Statistical Considerations

Statistical analysis was performed with the MedCalc Statistical Software version 14.10.2 (MedCalc Software bvba, Ostend, Belgium; http://www.medcalc.org; 2014). We assumed as primary endpoint overall response rate (ORR), defined as the proportion of patients who achieved a complete or partial response according to the Response Evaluation Criteria in Solid Tumours (RECIST) criteria version 1.0 [52]. Secondary endpoints were PFS and OS. PFS was defined as the interval between the treatment start date and death, first sign of clinical progression or last follow-up visit for patients lost at follow-up. OS was defined as the interval between the treatment start date and death or last follow-up visit for patients lost at follow-up. The association between categorical variables was estimated by Fisher exact test for binomial variables and by χ2 test for all other instances. Survival distribution was estimated by the Kaplan–Meier method. Significant differences in probability of survival between the strata were evaluated by log-rank test. For all statistical analysis, level of statistical significance alpha was set at 0.05.

### 4.3. Analysis of Serum Pro-Angiogenic Factors

Patients with at least one blood sample available were included. Although blood samples were collected before all cycles of chemotherapy, we planned to compare the previously identified neo-angiogenesis markers’ concentrations at three specifically defined time points.

The first blood sample (A) was taken just before starting the first cycle of chemotherapy. The second blood sample (B) was taken approximately two weeks after the first cycle (when the second cycle of chemotherapy is usually supposed to be done). Finally, the third blood sample (C) was taken just before the scheduled CT scan (usually at the fourth cycle of chemotherapy, around the eighth week of treatment).

Blood samples were performed before any kind of infusion (even before pre-medication for the scheduled cycle of chemotherapy).

We hypothesized that these three blood samples were the best to define the likelihood of treatment response. Blood sample A might be related to overall tumor burden, taking a strong “prognostic” role. Median and first and third quartile were used as cutoff points.

Concentrations’ difference between blood sample A/B should allow identifying early “responders” patients to treatment. This was defined as ratio between the samples A vs. B explored biomarker concentrations. The cutoff points were based upon the values of 25th and 75th percentiles (if the distribution was proven to be normal). Conversely, concentrations’ difference between blood sample A/C should identify the most representative “driver” of the tumor behavior. This was defined as ratio between the samples A and C explored biomarker concentration. The cutoff points were based upon the values of 25th and 75th percentiles (if the distribution was proven to be normal).

Additional blood samples were performed for patients who did not experience disease progression (PD) at their first radiological evaluation, until first radiological PD sign.

Serum levels of the following pro-angiogenic factors were analyzed: HGF, SDF-1, PlGF, FGF-2, MCP-3, IL-8, and VEGF. Frozen serum samples were shipped and analyzed centrally at the Medical Oncology Unit, University Hospital, and Università Politecnica delle Marche, in Ancona, Italy. Evaluation was performed by an ELISA-based technique.

The kit that was used to assess the different concentrations of FGF-2 (ELISA Fibroblast Growth Factor2 Kit (Cloud-clone Corp, product no. CEA551Hu, Segrate (MI), Italy), as stated in the homepage of the manufacturer (http://www.cloud-clone.com/products/CEA551Hu.html), has high specificity against FGF-2 and no relevant cross-reactivity or interference was observed for analogues of fibroblast growth factor 2. Kit manufacturer specifications were as follows: Intra-assay precision (precision within an assay): Three samples with low-, middle-, and high-level FGF-2, basic (FGF-2) were tested 20 times on one plate, respectively. Inter-assay precision (precision between assays): Three samples with low-, middle-, and high-level FGF-2, basic (FGF-2) were tested on three different plates, eight replicates in each plate. Coefficient of Variability (CV) (%) = SD/meanX100; intra-assay, CV < 10%; inter-assay, CV < 12%. The detection rate range of the kit was 12.35–1000 pg/mL, with a minimum detectable dose of less than 5.16 pg/mL.

### 4.4. Data Availability

The datasets used and/or analyzed during the current study are available from the corresponding author on reasonable request.

### 4.5. Statement of Ethics

This study was performed in accordance with the study protocol and the ethical principles stated in the Declaration of Helsinki as well as those indicated in the International Conference on Harmonization (ICH) Note for Guidance on Good Clinical Practice (GCP; ICH E6, 1995) and all applicable regulatory requirements. All patients had to sign a written informed consent before study entry. Adequate information was given to eligible patients by the principal investigator or co-investigators at each participating Center and in accordance with local regulations. Written informed consent to participate in the clinical study had to be given before any study-related activities were carried out. The declaration of informed consent was personally signed and dated by the subject and by the investigator/person designated by the investigator to conduct the informed consent discussion.

## 5. Conclusions

Our data suggest that FGF2 levels’ variations could be used as a potential biomarker in patients who are more likely to gain benefit from bevacizumab (and to a greater extent) and other anti-VEGF drugs. Further studies highlighting the FGF2 role during bevacizumab treatment are needed, in order to confirm its prognostic role. Interestingly, we found that FGF2 levels reach their highest value just before disease progression during bevacizumab treatment. These results might open new leads of research focused on FGF2 inhibition in second-line or more, for patients who are pretreated with anti-angiogenic-based therapy.

## Figures and Tables

**Figure 1 cancers-12-01330-f001:**
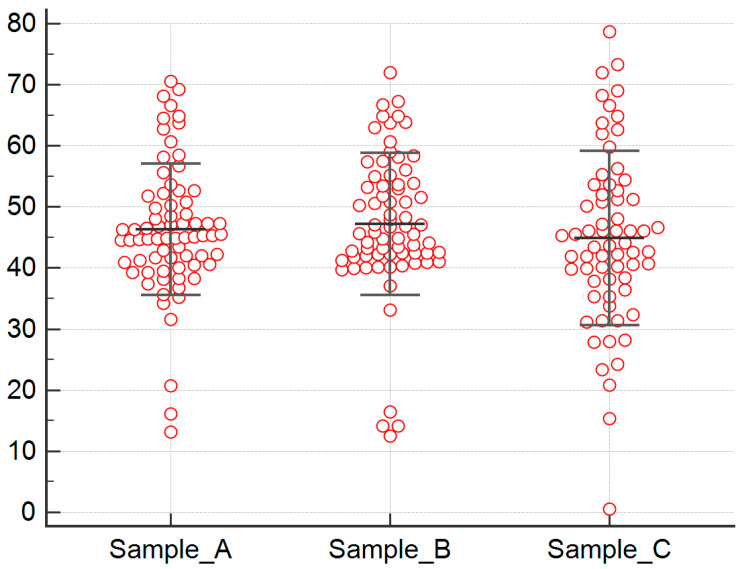
Dot plot of FGF-2 values for blood samples A, B, and C. High inter-patient variability for FGF-2 plasma levels at baseline (sample A), after two weeks (sample B), and at eight weeks (sample C). Blood sample A median FGF-2 serum level was 44.963 pg/mL (range: 13.203–73.618) and mean FGF-2 serum level was 46.355 pg/mL (standard deviation, SD: 11.060). Blood sample B median FGF-2 serum level was 44.919 (range 12.537–72) and mean FGF-2 serum level was 47.103 (SD: 11.763). Similarly, in blood sample C, median FGF-2 serum level was 43.509 (range 0.544–78.735) and mean FGF-2 serum level was 44.582 (SD: 14.141).

**Figure 2 cancers-12-01330-f002:**
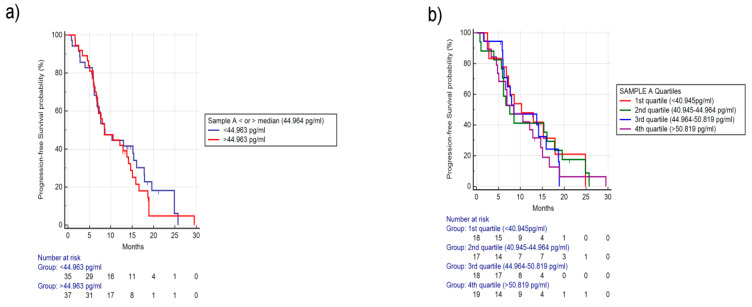
Progression Free Survival (PFS) based on baseline FGF-2 levels. (**a**) PFS stratified by median FGF-2, median Progression Free Survival (mPFS), respectively, for higher or lower than median: 8.52 vs. 8.60 months, Hazard Ratio (HR): 1.16, 95% Confidence Interval (CI): 0.70–1.92, *p* = 0.53). (**b**) PFS stratified in quartiles (first quartile: <40.945 pg/mL, second quartile: 40.945–44.964 pg/mL, third quartile: 44.945–50.819 pg/mL, fourth quartile >50.819 pg/mL), respectively, 10.3 vs. 7.54 vs. 8.52 vs. 8.49 months, *p* = 0.90.

**Figure 3 cancers-12-01330-f003:**
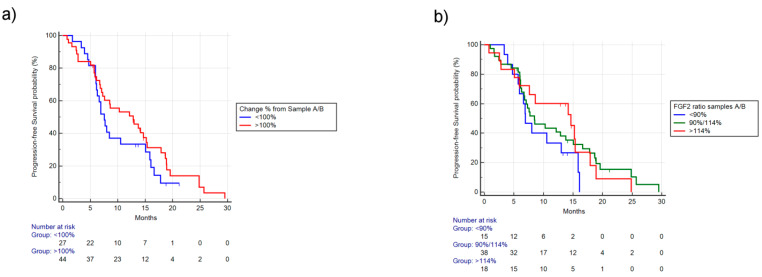
PFS based on ratio of FGF-2 levels between sample A/B. (**a**) PFS stratified by reduction (<100% concentration of FGF-2 between sample A/B) vs. increase (>100% concentration of FGF-2 between sample A/B). The mPFS for increase vs. reduction was, respectively, 12.85 vs. 7.57 months, HR: 0.73, 95% CI: 0.43–1.27, *p* = 0.23. (**b**) PFS stratified by different FGF-2 concentration ratios between sample A/B. The 10th percentile was 80%, the 25th percentile was 90%, the 75th percentile was 114%, and the 90th percentile was 123%. Using the 25th and 75th percentiles, we identified three groups (<90%, 90%–114%, >114% ratio between A/B), mPFS, respectively, was 6.95 vs. 8.49 vs. 14.66 months, *p* = 0.32.

**Figure 4 cancers-12-01330-f004:**
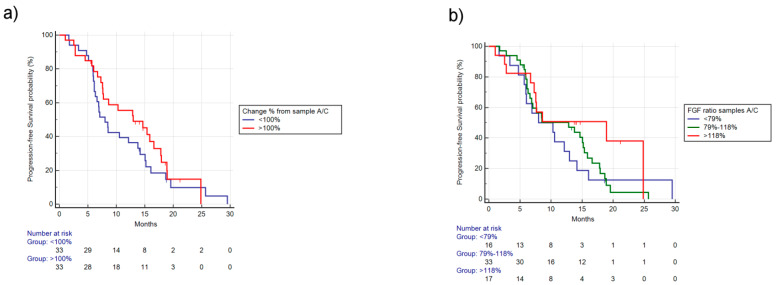
PFS based on ratio of FGF-2 levels between sample A/C. (**a**) PFS stratified by reduction (<100% concentration of FGF-2 between sample A/C) vs. increase (>100% concentration of FGF-2 between sample A/C). The mPFS for increase vs. reduction was, respectively, 12.98 vs. 8 months, HR: 0.78, 95% CI: 0.46–1.33, *p* = 0.35. (**b**) PFS stratified by different FGF-2 concentration ratio between sample A/C. The 10th percentile was 59%, the 25th percentile was 79%, the 75th percentile was 118%, and the 90th percentile was 137%. Using as cutoff the 25th and 75th percentiles, we identified three groups (<79%, 79%–118%, >118% ratio between A/C), mPFS was, respectively, 8.00 vs. 12.85 vs. 18.91 months, *p* = 0.35.

**Figure 5 cancers-12-01330-f005:**
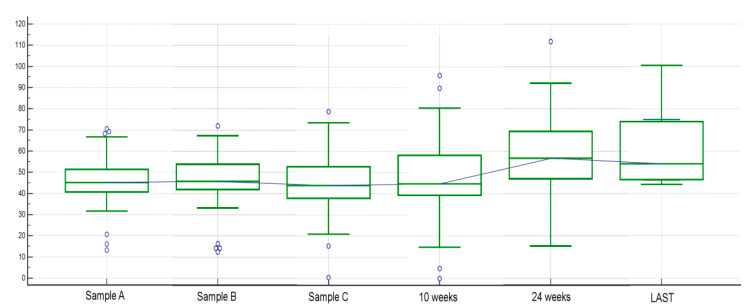
Circulating FGF-2 levels’ changes throughout various time points. FGF-2 levels’ concentration (pg/mL) in blood samples at baseline (sample A), two weeks (sample B), eight weeks (sample C), 10 weeks, 24 weeks, and, for patients who did not progress prior, sample at PD time (sample LAST).

**Figure 6 cancers-12-01330-f006:**
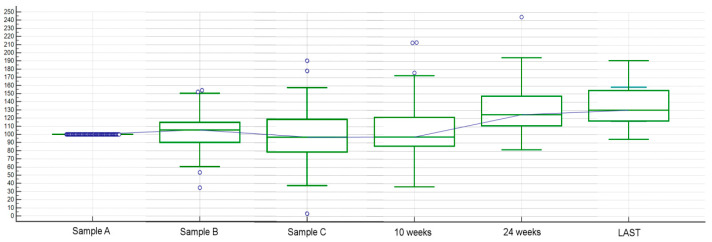
Circulating FGF-2 levels’ changes throughout various time points. FGF-2 levels’ percentage change (ratio) between baseline levels (sample A) and two weeks (sample B), eight weeks (sample C), 10 weeks, 24 weeks and, for patients who did not progress prior, sample at PD time (sample LAST).

**Figure 7 cancers-12-01330-f007:**
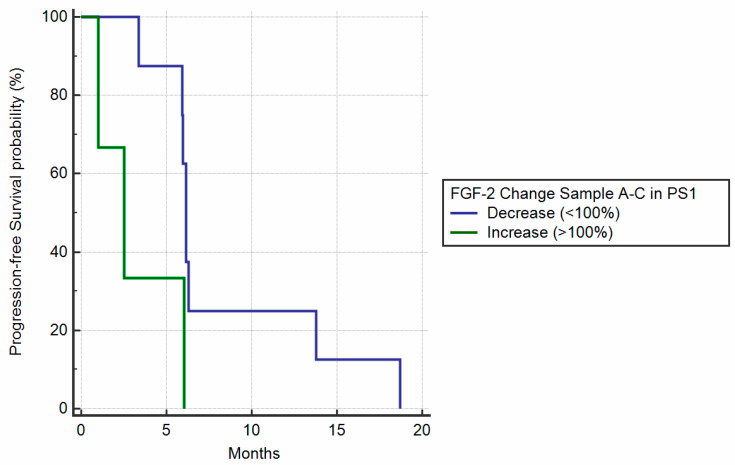
PFS in symptomatic (ECOG PS-1) patients stratified by increase of FGF-2 between baseline and eight weeks sample. Patients with symptomatic disease (ECOG PS-1) with FGF-2 levels’ increase between baseline and eight weeks’ treatment had a statistically significantly improved mPFS (6.13 vs. 2.52 months, HR: 3.67, *p* = 0.0294).

**Table 1 cancers-12-01330-t001:** Patients’ clinical characteristics.

Gender	41 (57%) Males 31 (43%) Females
Age	60 Y.O. (Median) 33–76 Y.O. (Range) 19 (26%) > 70 Y.O.
Sidedness	39 (54%) right-32 (46%) left sided (14 rectal cancer and 18 sigmoid cancer)
ECOG PS at treatment start	59 (82%) ECOG PS: 0 13 (18%) ECOG PS: 1
Synchronous/Metachronous metastatic involvement	39 (54%) had synchronous metastatic involvement 33 (46%) had metachronous metastatic involvement
LDH value	55 (76%) had LDH value lower than 1.17xULN (as per CENTRAL) 17 (24%) had LDH value higher than 1.17xULN (as per CENTRAL)
K-RAS/B-RAF/N-RAS status	42 (58%) patients were K-RAS mutant 25 (35%) patients were RAS/B-RAF wild type 4 (5%) patients were B-RAF mutant 1 (1%) patient was N-RAF mutant
Severe (>G3 NCI CTCAE) toxicities under treatment	58 (80%) did not experience G3 NCI CTCAE toxicities or higher 14 (20%) did experience G3 NCI CTCAE toxicities or higher

Patients’ clinical characteristics. Y.O.: years old; UNL: upper normal limit; NCI CTCAE: National Cancer Institute-Common Terminology Criteria for Adverse; ECOG PS: Performance Status according to Eastern Cooperative Oncology Group scale.

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
