# Peer review of "From CENTRAL to SENTRAL (SErum aNgiogenesis cenTRAL): Circulating Predictive Biomarkers to Anti-VEGFR Therapy"

_cancers, 2020, doi:10.3390/cancers12051330_

Round 1
Reviewer 1 Report
The study is a small prospective study looking at the changing levels of various angiogeneic factors in the treatment of patients with metastatic colorectal cancer. The correlation of the levels of FGF2 is shown to have some relatedness to patient outcome, but the link to the development of resistance to the anti-angiogenic therapy, rather than the combination therapy or the chemotherapy alone is not clear. I have the following comments:- 1. Some greater explanation needs to be given about the rationale for the selection of the biomarkers analysed. 2. Please explain why the biomarkers VEGF-C and VEGF-D were not analysed in this study given that they are alternative ligands for VEGFR-2 and have been previously shown to be upregulated in a number of tumour contexts. 3. What is the specificity of the FGF2 assay for FGF-2?, are other members of the broader FGF family of growth factors detected in this assay system? There are some good biological studies linking the VEGF and FGF signalling pathways in endothelial biology, this should be referred to in the text. 4. The blood samples were taken “upon” patients first treatment cycle. Does this language indicate that the sample was taken before that cycle, during or immediately after the cycle of treatment, this is not clear to me. I am not a clinically trained person but the terminology lacks a precision. As the treatments have specific chemotherapy combined the exact point of the blood sample relative to the treatment cycle is critical to the interpretation of the data. 5. It would be helpful if the mean and variance (or median and range) of the three groups in Figure 1 could be presented on the graphs for comparison. 6. What is known about the timeframe for the development of therapy resistance in these patients? Does it occur rapidly, by sample B, take longer, so might imply sample C or does it in fact take much longer and therefore might only be partially developed when samples B and C are taken? Figure 5 suggests that the timing of samples B and C might be less effective, this needs some further discussion.Author Response
English language and style
(x) Moderate English changes required
Response: thank you for your observation. We revised the English of the manuscript.
Comments and Suggestions for Authors
The study is a small prospective study looking at the changing levels of various angiogeneic factors in the treatment of patients with metastatic colorectal cancer. The correlation of the levels of FGF2 is shown to have some relatedness to patient outcome, but the link to the development of resistance to the anti-angiogenic therapy, rather than the combination therapy or the chemotherapy alone is not clear. I have the following comments
Point 1: Some greater explanation needs to be given about the rationale for the selection of the biomarkers analysed.
Response 1: Thank you for your suggestion. We try to explain the correlation between circulating proangiogenic factors levels and resistance to anti-angiogenic drugs. Two resistance mechanisms have been proposed to explain the angiogenesis inhibition: intrinsic and adaptive. Intrinsic resistance exists in tumours prior to treatment, while adaptive resistance arises after an initial response to antiangiogenic therapy. For both modalities, resistance may develop through compensatory pathways that mitigate the need for VEGF-mediated survival signalling. Among these mechanisms, we report activation and/or upregulation of alternative pro-angiogenic signalling pathways, recruitment of bone marrow-derived pro-angiogenic cells, increase pericyte coverage of the tumour vasculature and increased metastatic spread to provide access to normal tissue vasculature. (Bergers G. et al; Nat Rev Cancer 2008). Responsible for activating some of these mechanisms are both cancer cells but also anti-angiogenic therapy itself, which can evoke escape mechanisms inducing tumour vascularization. In particular, several preclinical studies demonstrated that the angiogenesis blockade in tumour-bearing mice upregulates expression of placental growth factor (PlGF), VEGF, fibroblast growth factor 2 (FGF2), and other angiogenic factors (Casanovas O et al. Cancer Cell 2005; Mitsuhashi A. et al. Nat. Commun 2015). Furthermore, in glioblastoma patients, plasma levels of FGF-2 and stromal-derived factor 1 (SDF-1) are increased upon disease progression under VEGF-targeted therapy (Batchelor TT et al. Cancer Cell 2007). Also in glioblastoma patients, Hepatocyte Growth Factor (HGF)/Tyrosine Protein Kinase Met (c-MET) pathway was highly upregulated during the recurrence after bevacizumab treatment. This phenomenon is hypoxia-dependent and it has not been recorded in patients not treated with anti-VEGF (Jahangiri A. et al. Clin Cancer Res 2013; Lu KV. et al. Cancer Cell 2012). Human Monocyte Chemotactic Protein-3 (MCP-3), is often produced by tumour cell lines. Its production might contribute to cancer cells invasion and metastasis, regulating protease secretion by macrophages (Opdenakker G. et al. Biochem Biophys Res Commun. 1993). Finally, certain inflammatory cytokines might have potent proangiogenic effects. In particular, interleukin-8 (IL-8) is a pro-angiogenic factor produced by tumour-infiltrating macrophages that has been revealed to facilitate the development of angiogenesis in various cancers. In xenograft cancer models, anti-angiogenic resistance coincided with increased secretion of IL-8 from tumour cells to the plasma (Dan Huang et al. Cancer Res February 1 2010).
Release of these angiogenic factors (VEGF-A, HGF, SDF1, PlGF, FGF-2, MCP-3 and IL-8), induced by hypoxia in the tumour tissue, may be associated with anti-VEGF resistance. Monitoring their serum levels can be a surrogate predictive biomarker of anti-angiogenic therapies.
This explanation has been added in the introduction section (lines 72-87).
Point 2: Please explain why the biomarkers VEGF-C and VEGF-D were not analysed in this study given that they are alternative ligands for VEGFR-2 and have been previously shown to be upregulated in a number of tumour contexts.
Response 2: VEGF-C and VEGF-D are both involved in lymphangiogenesis, due to the high affinity for VEGFR-3, which is expressed on lymphatic endothelial cells. They have a weak affinity for VEGFR-2, activating the angiogenesis occasionally. To date, their role in angiogenesis remains controversial. Some data in the literature support that VEGF-C has been associated with angiogenesis in breast cancer and showed that it works with FGF2 and VEGF-A in synergy in order to induct angiogenesis. Another study suggested that VEGF-C induces blood vessel changes without evidence of new angiogenesis. Few data report arguments on the role of VEGF-D and angiogenesis, but a study of patients with CRC found that lower expression of VEGF-D was associated with greater benefit from bevacizumab treatment. In our study, we selected proangiogenic biomarkers, with particular interest on those that showed more correlations with antiangiogenic drugs resistance. Our propose was to elaborate a clinical background to expand knowledge about other proangiogenic factors as VEGF-C and VEGF-D.
Point 3: What is the specificity of the FGF2 assay for FGF-2? are other members of the broader FGF family of growth factors detected in this assay system? There are some good biological studies linking the VEGF and FGF signalling pathways in endothelial biology, this should be referred to in the text.
Response 3: The Kit that was used to assess the different concentrations of FGF-2 (ELISA Fibroblast Growth Factor2 Kit (Cloud-clone Corp, product no. CEA551Hu), as stated in the homepage of the manufacturer (http://www.cloud-clone.com/products/CEA551Hu.html) has high specificity against FGF-2 and no relevant cross-reactivity or interference has been observed for analogues of fibroblast growth factor 2. This has been added to the methods section (lines 406-410).
Point 4: The blood samples were taken “upon” patients first treatment cycle. Does this language indicate that the sample was taken before that cycle, during or immediately after the cycle of treatment, this is not clear to me. I am not a clinically trained person but the terminology lacks a precision. As the treatments have specific chemotherapy combined the exact point of the blood sample relative to the treatment cycle is critical to the interpretation of the data.
Response 4: Thank You for your comment. Blood samples were performed before any kind of infusion (even before pre-medication for the scheduled cycle of chemotherapy).
The first blood sample (A) FOLFIRI + bevacizumab regimen was taken just before starting the first cycle of chemotherapy. The second blood sample (B) was taken approximately 2 weeks after the first cycle (when the 2nd cycle of chemotherapy is usually supposed to be done). Finally, the third blood sample (C) was taken just before the scheduled CT scan (usually at the 4th cycle of chemotherapy, around the 8th week of treatment). This has been added in the methods section (lines 381-389).
Point 5: It would be helpful if the mean and variance (or median and range) of the three groups in Figure 1 could be presented on the graphs for comparison.
Response 5: We added in Figure 1 markers and brackets to identify mean and SD of the data distribution of samples
Point 6: What is known about the timeframe for the development of therapy resistance in these patients? Does it occur rapidly, by sample B, take longer, so might imply sample C or does it in fact take much longer and therefore might only be partially developed when samples B and C are taken? Figure 5 suggests that the timing of samples B and C might be less effective, this needs some further discussion.
Response 6: Thank You for the suggestion. FOLFIRI+Bevacizumab has been extensively used in metastatic colorectal cancer. Consequently, it is well known the proportion of patients who develop resistance, during the treatment. For example, the FIRE-3 trial randomized mCRC (k-RAS wild type) patients to receive FOLFIRI + Bevacizumab or FOLFIRI + cetuximab as first line. In FOLFIRI + bevacizumab arm, median PFS is around 10 months and at 8 weeks-timepoint the 7% of patients had experienced disease progression. In our study, median PFS is around 12 months, similar to FIRE-3 populations. However, earlier detection of resistance (for example at timepoint 2 weeks), is actually unknown. Therefore, we can’t answer to your question whether some surrogate of early detection of resistance does exist. This is why we conducted our study.

Reviewer 2 Report
The paper investigated serum pro-angiogenic factors associated with prognosis of metastatic colorectal cancer (mCRC) patients treated with bevacizumab, an antibody directed against soluble circulating vascular endothelial growth factor-A (VEGF-A). Author found early increase of circulating levels of FGF-2 could be used as a marker to identify patients who are more likely to gain benefit from bevacizumab first-line therapy. However, critical issues needed to be addressed as follows.
Comments and Questions:
1. In the discussion, author claim low baseline circulating FGF2 levels seem to suggest an improved prognosis. This is not consistent with the result described in 2.1.1. (Fig. 2A, B) demonstrating that no outcome differences were found considering different baseline FGF2 concentrations for mPFS and mOS
2. Author argued that the FGF2 levels increase during bevacizumab treatment might be associated with better survival outcomes. However, the underlying mechanisms are hard to be defined. Two critical issues need to be addressed
i. While author suggest that patients under bevacizumab treatment may show a FGF2 increase that could potentially explain the secondary resistance (line 245-251), they didn’t explain why increased FGF2 was associated improved prognosis in their results.
ii. Author suggested that a FGF2 levels decrease might be related to an ineffective inhibition of VEGF-A. However, they didn’t demonstrate an obious correlation between VEGF-A and clinical outcome (and FGF2 levels) during bevacizumab treatment .
3. In vitro study needed to be performed to examine whether mCRC cell treated with bevacizumab may result in compensatory enhance of FGF2
4. In vivo study needed to be performed to validate whether bevacizumab treatment may increase FGF2 in animals with mCRC that correlate with a better outcome.
Author Response
Dear Reviewer,
Thank you for your comments. Here are the answers to your questions.
English language and style
(x) English language and style are fine/minor spell check required
Response: thank you for your observation. We revised English language and style.
Comments and Suggestions for Authors
The paper investigated serum pro-angiogenic factors associated with prognosis of metastatic colorectal cancer (mCRC) patients treated with bevacizumab, an antibody directed against soluble circulating vascular endothelial growth factor-A (VEGF-A). Author found early increase of circulating levels of FGF-2 could be used as a marker to identify patients who are more likely to gain benefit from bevacizumab first-line therapy. However, critical issues needed to be addressed as follows.
Comments and Questions:
Point 1: In the discussion, author claim low baseline circulating FGF2 levels seem to suggest an improved prognosis. This is not consistent with the result described in 2.1.1. (Fig. 2A, B) demonstrating that no outcome differences were found considering different baseline FGF2 concentrations for mPFS and mOS
Response 1: Thank You for your suggestions. Probably, we didn't explain ourselves well. In the results section, we highlight that no outcome differences were found considering different baseline FGF2 concentrations. In particular, no differences in term of mPFS, mOS an RR in patients with higher vs patient with lower median FGF2 level (Fig 2A). Furthermore, stratifying patients in quartiles (1st quartile: <40.945 pg/ml, 2nd quartile: 40.945-44.964 pg/ml, 3rd quartile: 44.945-50.819 pg/ml, 4th quartile >50.819 pg/ml), no statistically significant differences were seen among the 4 groups in term of PFS (Figure-2b) and OS. However, patients included in the 4th quartile had a trend towards worse OS (mOS respectively NR vs 24.85 vs NR vs 20.75 months, for 1st vs 2nd vs 3rd vs 4th quartile, p=0.47).
This result, although not statistically significant, suggests that low baseline circulating FGF2 levels appear to be associated with an improved prognosis. This explanation has been added in the discussion section (line 251-254).
Point 2: Author argued that the FGF2 levels increase during bevacizumab treatment might be associated with better survival outcomes. However, the underlying mechanisms are hard to be defined. Two critical issues need to be addressed
i) While author suggest that patients under bevacizumab treatment may show a FGF2 increase that could potentially explain the secondary resistance (line 245-251), they didn’t explain why increased FGF2 was associated improved prognosis in their results.
ii) Author suggested that a FGF2 levels decrease might be related to an ineffective inhibition of VEGF-A. However, they didn’t demonstrate an obious correlation between VEGF-A and clinical outcome (and FGF2 levels) during bevacizumab treatment.
Response 2: Thank You for your comments. In our study, the circulating FGF2 serum levels may correlate with clinical outcome for patients treated with FOLFIRI/bevacizumab. While low baseline circulating FGF2 levels seem to suggest an improved prognosis, the FGF2 levels increase during treatment might be associated with better survival outcomes.
A clear explanation to this biological phenomenon is difficult to find.
A probable explanation could be linked to the FGF2 pathway activation as a mechanism of resistance to antiangiogenic therapy. Therefore, a blockade of VEGF-A, mediated by bevacizumab may induce a rapid activation of FGF2, allowing escape from anti-angiogenic treatment.
Several studies showed the prognostic role of circulating proangiogenic factor, in CRC. Kwon et al analysing serum samples from 132 CRC patients undergoing curative resection. They showed that high preoperative VEGF levels were associated with increased tumour size and higher CEA levels, and in multivariate analysis, high VEGF‐A was an independent prognostic factor for shorter OS (HR 4.779, 95% CI 1.15–19.94, p = 0.032) (Kwon KA et al. BMC Cancer 2010).
Afterwards, in a retrospective subset analysis of more than 2,000 mCRC patients, high baseline levels of serum/plasma VEGF‐A and CEA were confirmed as prognostic biomarkers for poorer PFS and OS in patients with mCRC, independent of treatment (Jurgensmeier JM et al. Br J Cancer 2013).
In the discussion section, we speculate that a FGF2 levels decrease might be related to an ineffective inhibition of VEGF-A. Consequently, tumour cells would not induce the other pro-angiogenic factors production, because VEGF-A is already sufficient to sustain the neo-angiogenetic process.
As described in discussion section, the only caveat against this explanation is that our analysis failed to demonstrate a correlation between VEGF levels and FGF2.
Notably, most of these proangiogenic factors (VEGF-A and PlGF) are stored inside platelets. Platelet‐derived angiogenic cytokines are released from activated platelets and particularly from EDTA‐activated platelets. There is a large variation in susceptibility to platelet EDTA activation among individuals. Thus, their release by platelets could reduce the reliability of their circulating levels assessment as markers of different anti-angiogenic activity (Zimmermann R et al. J Immunol Methods 2009). Conversely, FGF2 is released by endothelial cells reducing the confounding factor of not using platelet-poor plasma, during blood samples processing.
This explanation has been added in the discussion section. (lines 258-267; lines 318-324)
Point 3: In vitro study needed to be performed to examine whether mCRC cell treated with bevacizumab may result in compensatory enhance of FGF
Point 4: In vivo study needed to be performed to validate whether bevacizumab treatment may increase FGF2 in animals with mCRC that correlate with a better outcome.
Response 3 and 4: Thank You for your comments. The role of FGF-2 in acquired resistance to anti-VEGF based therapy was already discussed in some pre-clinical studies.
Gyanchadani et al (Mol Cancer Res. 2013 Dec) demonstrated that, in HNC cell lines, acquired resistance to Bevacizumab treatment is associated with up-regulation of different pro-angiogenic factors. Among them FGF-2 plays a crucial role and has been found to be associated with restored sensitivity to Bevacizumab when inhibition of FGF-2 was performed in their model.
Ichikawa et al (Sci Rep. 2020) reported that, both in cell lines and in an animal model, that acquired resistance to anti-VEGF based therapy is associated with over-expression of both FGF-2 and FGFR2 in an almost specific manner. Tumour over-expressing VEGFR2-Fc showed highest levels of mRNA expression from FGF2 and other factors of FGF-2/FGFR2 pathway.
Guerrouehan et al (Mol Cancer Ther. 2014) demonstrated that, in ovarian cancer resistant to Bevacizumab, AKT-mediated endothelial factors secretion (where FGF-2 plays a major role) creates an autocrine loop that seemed intended to escape from Bevacizumab inhibition. Cell lines subject of this study, were obtained through a continuous exposure to Bevacizumab, fin order to facilitate the resistance model acquisition.
We added these data in the discussion section (lines 289-301). They could contribute to our findings concerning the increase of FGF-2 levels in patients with stable disease (SD) or partial response (PR) when treated with FOLFIRI+Bevacizumab.
According to a brief Literature review, our study seems to be one of the first suggesting that primary (and not acquired) Bevacizumab resistance could be associated with a reduction of the levels of circulating FGF2
Please see the attachement.

Round 2
Reviewer 1 Report
Thank you to the authors for addressing my comments.
I had two small requests.
For comment #2 please add a suitable mention of VEGF-C and VEGF-D in the text so that the readers are well aware of these alternative ligands to VEGF, and a suitable reference
For #5 can the mean and error bars be move to the side of the dot plots so they are easily visible, otherwise they will need to be thicker/bolder which might impeed the dots.
Author Response
Dear Reviewer,
Thank You for your comments.
Below our answers.
English language and style
Moderate English changes required
Response: thank you for your observation. We revised the English of the manuscript.
Comments and Suggestions for Authors
Thank you to the authors for addressing my comments.
I had two small requests.
For comment #2 please add a suitable mention of VEGF-C and VEGF-D in the text so that the readers are well aware of these alternative ligands to VEGF, and a suitable reference
Response: Thank You for your comments. VEGF-C and VEGF-D are involved in lymphangiogenesis, due to the high affinity for VEGFR-3 expressed on lymphatic endothelial cells (Joukov V et al, EMBO J. 1996). Their poor affinity for VEGFR-2 activates the angiogenesis occasionally. To date, their role in angiogenesis remains controversial. The literature data support that VEGF-C has been associated with angiogenesis in breast cancer showing its cooperation with FGF2 and VEGF-A in order to induct angiogenesis. Another study suggested that VEGF-C induces blood vessel changes without evidence of new angiogenesis (Tille JC et al, J Pharmacol Exp Ther. 2001; Valtola R et al, Am J Pathol. 1999; Benest AV et al, Cardiovascular Research 2008). Few data reports comment on the role of VEGF-D and angiogenesis, but a study over CRC patients found that lower expression of VEGF-D was associated with an outcome benefit from bevacizumab treatment (Lieu CH et al, PLoS One. 2013; Weickhardt AJ et al, Br J Cancer. 2015).
We added the part relating the alternative ligands in the introduction section (lines 87-106)
For #5 can the mean and error bars be move to the side of the dot plots so they are easily visible, otherwise they will need to be thicker/bolder which might impeed the dots.
Response: We modified Figure 1 as suggested.
Please see the attachment.

Reviewer 2 Report
In the revised version, author provide informations regarding the issues raised in my comments. Part of them is meaningful. For example, they mentioned most proangiogenic factors (VEGF-A and PlGF) are stored inside platelets and may be released from from EDTA‐activated platelets in whole blood sample, whereas FGF2, released by endothelial cells, may not cause such variation.
However, several concerns still remain as following.
- In Response 2, author provide a lot of informations concerning the role of FGF-2 in acquired resistance to anti-VEGF based therapy in some pre-clinical studies (line 281-293), however, the controversy why increased FGF2 was associated improved prognosis in their results cannot be addressed as before. Probably this controversy is due to the profound differences of biological/ microenviromental context existing between preclinical and clinical trial. Author needs to explain this more comprehensively
- In Response 1, in order to justify that low baseline circulating FGF2 levels appear to be associated with an improved prognosis, author indicate that patients included in the 4th quartile had a trend towards worse OS (mOS respectively NR vs 24.85 vs NR vs 20.75 months, for 1st vs 2nd vs 3rd vs 4th quartile, p=0.47). Although this finding is interesting, however, it is not statistically significant. Author need to increase the sample numbers in order to improve the statistic.
Author Response
Dear Reviewer,
Thank You for your comments. Here below our answers.
English language and style
English language and style are fine/minor spell check required
Response: thank you for your observation. We revised the English of the manuscript.
Comments and Suggestions for Authors
In the revised version, author provide informations regarding the issues raised in my comments. Part of them is meaningful. For example, they mentioned most proangiogenic factors (VEGF-A and PlGF) are stored inside platelets and may be released from from EDTA‐activated platelets in whole blood sample, whereas FGF2, released by endothelial cells, may not cause such variation. However, several concerns still remain as following.
- In Response 2, author provide a lot of informations concerning the role of FGF-2 in acquired resistance to anti-VEGF based therapy in some pre-clinical studies (line 281-293), however, the controversy why increased FGF2 was associated improved prognosis in their results cannot be addressed as before. Probably this controversy is due to the profound differences of biological/ microenviromental context existing between preclinical and clinical trial. Author needs to explain this more comprehensively
Response: Thank You for your suggestion.
In Literature the prognostic impact of circulating FGF2 levels is still controversial. Serum FGF2 levels showed prognostic significance in several clinical trial (Akl MR et al, Oncotarget. 2016.). Therefore, dosing FGF2 serum levels may provide an indirect, non-invasive way to monitor progression cancer.
Increased FGF2 levels have been found in several human cancers, but FGF2 levels do not always correlate with microvessel density (Hoying JB et al, J Cell Physiol. 1996; Presta M et al, Cytokine Growth Factor Rev 2005). In fact, FGF2 contributes to tumour progression not only by inducing neovascularizzation but also by modulating growth, differentiation, migration, and survival in a variety of cancer cell types (Haugsten EM et al, Mol Cancer Res. 2010; Casanovas O et al, Cancer Cell 2005). Several studies have shown that FGF2 is a key cancer-promoting factor in the tumour microenvironment, regulating cross-talk between epithelial and stromal tumour compartment. In particular, FGF2 expression is high in the tumour stroma, including inflammatory cells, myofibroblasts, and endothelial cell, suggesting that FGF2 can modulate tumour progression (Pietras K, PLoS Med. 2008). This may partially explain the difficulty in demonstrating the prognostic role of FGF2 in preclinical studies. Cancer cell lines in vitro and in vivo trial do not totally represent microenvironment heterogeneity and complexity (Sulaiman A et al, Oncotarget. 2017; Prasetyanti PR et al, Mol Cancer. 2017). This explanation has been added in the discussion section (Lines 382-391).
- In Response 1, in order to justify that low baseline circulating FGF2 levels appear to be associated with an improved prognosis, author indicate that patients included in the 4th quartile had a trend towards worse OS (mOS respectively NR vs 24.85 vs NR vs 20.75 months, for 1st vs 2nd vs 3rd vs 4th quartile, p=0.47). Although this finding is interesting, however, it is not statistically significant. Author need to increase the sample numbers in order to improve the statistic.
Response: Thank You for your comment. The SENTRAL study (SErum aNgiogenesis-cenTRAL) is an exploratory pre-planned analysis in phase II CENTRAL trial, evaluating first-line FOLFIRI and bevacizumab in patients with advanced colorectal cancer, prospectively stratified according to serum LDH. As the CENTRAL trial is closed, it is difficult to increase the sample size. The SENTRAL aim was to identify concentration changes of circulating pro-angiogenic factors during treatment as a potential prognostic or predictive factor for efficacy/resistance to FOLFIRI/bevacizumab treatment. Although not statistically significant, our analysis suggests that the circulating FGF2 serum levels may correlate with clinical outcome for patients treated with FOLFIRI/bevacizumab.
Therefore, our analysis lays the foundation for future, larges prospective investigations on the FGF2 role.
Please see the attachment.
